# Q-LEARNING PENALIZED TRANSFORMER FOR SAFE OFFLINE REINFORCEMENT LEARNING

## ABSTRACT

This paper addresses the problem of safe offline reinforcement learning, which involves training a policy to satisfy safety constraints using an offline dataset. This problem is inherently challenging as it requires balancing three highly interconnected and competing objectives: satisfying safety constraints, maximizing rewards, and adhering to the behavior regularization imposed by the offline dataset. To tackle this trilogy challenge, we propose a novel framework, the Q-learning Penalized Transformer policy (QPT). Specifically, QPT adopts a sequence modeling paradigm, learning the action distribution conditioned on historical trajectories and target returns, thereby ensuring robust behavior regularization. Additionally, we incorporate Q-learning penalization into the training process to optimize the policy by maximizing the expected reward and minimizing the expected cost, guided by the learned Q-networks. Theoretical analysis demonstrates the advantages of our approach by aligning with optimal policies under mild assumptions. Experimental results across 38 tasks further validate the effectiveness of the QPT framework, demonstrating its ability to learn adaptive, safe, robust, and high-reward policies. Notably, QPT consistently outperforms strong safe offline RL baselines by a significant margin across all tasks. Furthermore, it retains zero-shot adaptation capabilities to varying constraint thresholds, making it particularly well-suited for real-world RL scenarios that operate under constraints.

## 1 INTRODUCTION

Offline reinforcement learning (RL) focuses on learning effective policies entirely from previously collected data, without requiring interaction with the environment (Fujimoto et al., 2019). This paradigm has emerged as a powerful approach for addressing sequential decision-making tasks, such as autonomous driving (Hu et al., 2022) and control systems (Zhan et al., 2022). Various paradigms have been developed to maximize the utility of pre-collected trajectories while mitigating policy overfitting (Kumar et al., 2019; Fujimoto et al., 2019; Kostrikov et al., 2021a; Kumar et al., 2020). However, standard offline RL often falls short in real-world applications, where diverse safety constraints limit feasible solutions, making the mere maximization of a scalar reward function insufficient. The requirement for safety, or the satisfaction of constraints, is particularly critical when deploying RL algorithms in real-world scenarios (García & Fernández, 2015). Ensuring constraint satisfaction not only broadens the applicability of RL methods but also enhances their reliability in safety-critical domains.

Developing an optimal policy within a constrained manifold has been a central focus of recent research in safe offline RL, which seeks to integrate safety requirements into offline RL frameworks (García & Fernández, 2015). Several approaches bridge concepts from offline RL and safe RL, employing techniques such as pessimistic estimations (Xu et al., 2022) and stationary distribution correction (Lee et al., 2022). Constrained optimization formulations, often incorporating Lagrange multipliers, are commonly used to identify policies that maximize rewards while adhering to safety constraints (Le et al., 2019). Sequential modeling methods, such as the Transformer (Liu et al., 2023b) and Diffuser (Lin et al., 2023; Zheng et al., 2024), have also been explored, demonstrating promising results in achieving both optimal policies and satisfying safety requirements.

However, the challenges of safe offline RL are amplified by the offline setting, which necessitates behavior regularization to mitigate distributional shift (Fujimoto et al., 2019). Balancing constraint

satisfaction, reward maximization, and offline policy regularization is particularly difficult due to the intricate inter-dependencies among these objectives. Jointly optimizing them often results in unstable training and suboptimal safety performance (Lee et al., 2022; Zheng et al., 2024). Furthermore, these objectives may inherently conflict (Xu et al., 2022). For instance, adhering to offline policy regularization can compromise constraint satisfaction when the dataset includes unsafe trajectories. Conversely, excluding unsafe trajectories may lead to suboptimal policies by omitting critical high-reward data, underscoring the inherent trade-offs in safe offline RL. Besides, Lagrange-based methods frequently integrate the constraint threshold as a constant within the training process, seeking to optimize policy performance while adhering to specified constraints (Le et al., 2019). We argue that the ability to adapt a trained policy to varying constraint thresholds is crucial for a wide range of real-world applications. In practice, enforcing stricter constraints typically results in diminished task performance and induces more conservative agent behaviors (Liu et al., 2022b). Consequently, our objective is to investigate a training paradigm that enables an agent to dynamically adjust its constraint threshold at deployment. This approach would allow for flexible control over the agent's level of conservativeness, eliminating the need for additional fine-tuning or retraining.

To address these challenges, we propose a novel safe offline RL approach, the Q-learning Penalized Transformer policy (QPT). Specifically, QPT adopts a sequence modeling paradigm that learns the action distribution conditioned on both historical trajectories and target returns, thereby enabling robust behavioral regularization and facilitating zero-shot adaptation to diverse deployment scenarios. Moreover, QPT employs separate reward and cost Q-networks, which are iteratively updated using the n-step Bellman equation. These networks are seamlessly integrated into the training process, where the conflicting objectives are formulated as a weighted combination of losses. This design provides explicit guidance for the policy to maximize expected rewards while effectively constraining expected costs, thereby promoting both safety and performance in offline RL settings. Theoretical analysis demonstrates the advantages of our approach by aligning with optimal policies under mild assumptions. Experimental evaluations on 38 tasks from the DSRL benchmark (Liu et al., 2023a) further validate QPT's effectiveness, showing its ability to learn safe, robust, and high-reward policies. QPT consistently outperforms state-of-the-art baselines across all tasks by a significant margin. Moreover, it retains zero-shot adaptation capabilities to varying constraint thresholds, making it particularly well-suited for real-world RL applications with safety requirements.

## 2 RELATED WORK

**Offline RL** trains policies from a static offline dataset $\mathcal{D}$, without online interaction (Levine et al., 2020), making it ideal for scenarios where interaction is costly or unsafe. A major challenge is *distribution shift*, where the learned policy deviates from the behavior policy, causing performance degradation (Fujimoto et al., 2019). To address this, prior works have employed constrained or regularized dynamic programming to limit policy deviations (Fujimoto & Gu, 2021; Kumar et al., 2020; Kostrikov et al., 2021b). Conditional sequence modeling predicts future actions from past experiences, constraining the policy within behavior boundaries and enabling zero-shot adaptability (Chen et al., 2021a; Hu et al., 2025; 2024d; Yamagata et al., 2023; Hu et al., 2023; 2024c).

**Safe RL** involves learning policies that maximize long-term rewards while satisfying safety constraints (Wachi & Sui, 2020; Gu et al., 2022). A common approach to constrained optimization in safe RL is the primal-dual framework, which reformulates the problem into an unconstrained optimization using Lagrangian multipliers (Chen et al., 2021b). Correction-based methods provide another solution by projecting unsafe actions onto safe sets, incorporating domain knowledge to improve exploration safety (Zhao et al., 2021; Luo & Ma, 2021). Model-based RL has also been applied to improve data efficiency and performance (Huang et al., 2023), though it often requires larger models to parameterize environment dynamics, increasing computational complexity.

**Safe offline RL** has also received growing attention, with the goal of ensuring zero constraint violations during inference. These methods combine offline RL techniques with safety constraints, such as using DICE-style approaches for constrained optimization (Polosky et al., 2022; Lee et al., 2022) and Lagrangian-based methods for simplicity and compatibility with existing offline RL frameworks. Recent studies have introduced novel networks for safe offline RL (Koirala et al., 2024b;a; Gong et al., 2025). For instance, CDT (Liu et al., 2023b) employs sequential modeling to learn from trajectory datasets, while TREBI (Lin et al., 2023) and FISOR (Zheng et al., 2024)

utilize diffusion models for safe policy development. TREBI generates safe trajectories directly, whereas FISOR uses a diffusion actor to constrain actions within feasible regions. In contrast to these approaches, we propose a novel framework that explicitly formulates the objectives as controllable losses, enabling the development of policies that directly align high rewards with minimal costs.

## 3 METHODOLOGY

### 3.1 PROBLEM SETUP

RL problems with safety constraints are naturally formulated within the Constrained Markov Decision Process (CMDP) framework (Altman, 1998). A CMDP is defined by a tuple $\mathcal{M} = (\mathcal{S}, \mathcal{A}, \mathcal{T}, \mathcal{R}, \mathcal{C}, \mu_0)$, where $\mathcal{S}$ is the state space, $\mathcal{A}$ is the action space, $\mathcal{T} : \mathcal{S} \times \mathcal{A} \times \mathcal{S} \rightarrow [0,1]$ is the state transition probability function, $\mathcal{R} : \mathcal{S} \times \mathcal{A} \rightarrow \mathbb{R}$ defines the reward function, and $\mathcal{C} : \mathcal{S} \times \mathcal{A} \rightarrow [0, C_{\max}]$ quantifies the costs associated state-action pairs, where $C_{\max}$ is the maximum possible cost, and $\mu_0 : \mathcal{S} \rightarrow [0,1]$ is the initial state distribution.

In safe offline RL problems, we are given a fixed pre-collected dataset $\mathcal{D}=\{(\mathbf{s}, \mathbf{a}, r, c, \mathbf{s}')_i\}_{i=1}^{\mathcal{H}}$ from one or more (unknown) behavior policies, where each training example $i$ contains the action $\mathbf{a}$ taken at state $\mathbf{s}$, reward received $r$, cost incurred $c$, and the next state $\mathbf{s}'$. The goal is to learn a policy $\pi : \mathcal{S} \rightarrow \mathcal{A}$ from the offline dataset $\mathcal{D}$ to maximize the expected reward while satisfying a specified cost/safety constraint. This problem is mathematically formulated as:

$$\max_{\pi} \mathbb{E}_{\tau \sim \pi}[R(\tau)] \quad \text{subject to} \quad \mathbb{E}_{\tau \sim \pi}[C(\tau)] \leq \kappa. \tag{1}$$

Here, $\kappa \in [0, +\infty)$ is the cost threshold for safety constraint, $\mathcal{H}$ is the horizon length of episode, $\tau = \{\mathbf{s}_1, \mathbf{a}_1, r_1, c_1, \ldots, \mathbf{s}_{\mathcal{H}}, \mathbf{a}_{\mathcal{H}}, r_{\mathcal{H}}, c_{\mathcal{H}}\}$ denotes a trajectory sampled by executing the policy $\pi$, $R(\tau) = \sum_{t=1}^{\mathcal{H}} r_t$ is the total accumulated reward, and $C(\tau) = \sum_{t=1}^{\mathcal{H}} c_t$ is the total incurred cost.

Most existing offline safe RL methods approach policy training as a constrained optimization problem, wherein learnable dual variables are updated according to estimates of constraint violation costs and a target threshold (Xu et al., 2022; Lee et al., 2022; Polosky et al., 2022). While this constrained optimization paradigm is effective in online safe RL settings (Stooke et al., 2020), it faces significant challenges in the offline context (Liu et al., 2023b). First, offline RL policies often become either unsafe or overly conservative due to biased value estimates, stemming from incomplete dataset coverage. In the Lagrangian dual optimization of Equation 1, such bias in cost estimation $C(\tau)$ can mislead dual variable updates relative to the fixed threshold $\kappa$, resulting in unsafe or overly cautious behaviors, a problem exacerbated in offline settings (Liu et al., 2022a). Second, policies cannot adapt to new constraint thresholds without retraining, as the threshold must remain fixed during training. Changing it post hoc destabilizes dual variables and can cause optimization to diverge, thus requiring full retraining for each new constraint.

To address these issues, we reformulate the learning objective described in Equation 1 and leverage sequential modeling techniques, which have shown promise in achieving zero-shot adaptation to varying constraint thresholds while maintaining near-optimal task performance (Hu et al., 2024b; Liu et al., 2023b). However, a key limitation of sequence modeling in this context is its tendency to imitate the behavior distribution present in the training dataset (Brandfonbrener et al., 2022), which often includes unsafe trajectories. Directly learning from such datasets may therefore result in unsafe policies. One potential remedy is to filter out unsafe trajectories from the dataset $\mathcal{D}$ to construct a "safe" dataset. Unfortunately, this strategy often eliminates high-reward transitions, leading to suboptimal policies. Ideally, the optimal solution would involve selectively "stitching" together transitions from both safe and unsafe trajectories, empowering the policy to generate behaviors that both maximize cumulative rewards and satisfy safety constraints. To this end, we propose a novel framework centered on the Conditional Transformer Policy (Section 3.2), augmented by Q-learning Penalization (Section 3.3), Data Augmentation, and an Ensemble Policy for the inference stage (Section 3.4). We also provide formal theoretical analysis to justify the observed performance gains of our methods (Section 3.5).

### 3.2 CONDITIONAL TRANSFORMER POLICY

The Transformer architecture (Vaswani et al., 2017), extensively studied in NLP (Devlin et al., 2018) and CV (Dosovitskiy et al., 2020), has also been explored in RL through the conditional sequence

modeling (CSM) paradigm (Hu et al., 2024b). In contrast to most traditional RL methods, which rely on value function estimation or policy gradient computation, DT (Chen et al., 2021a) directly predicts desired future actions based on a sequence of historical data comprising state ($\mathbf{s}_t$), action ($\mathbf{a}_t$), and return-to-go ($\hat{r}_t = \sum_{i=t}^{T} r_i$) tuples. In the context of safe offline RL, this formulation is extended by including an additional cost-to-go token, $\hat{c}_t = \sum_{i=t}^{T} c_i$, which quantifies the cumulative cost from the current time step to the end of the episode (Liu et al., 2023b). During training on offline data, the Transformer processes trajectory sequences in an auto-regressive manner, utilizing a historical context of the most recent $K$ steps. A trajectory sequence $\tau_t$ is formulated as follows:

$$\tau_t = (\hat{r}_{t-K+1}, \hat{c}_{t-K+1}, \mathbf{s}_{t-K+1}, \mathbf{a}_{t-K+1}, \ldots, \hat{r}_t, \hat{c}_t, \mathbf{s}_t, \mathbf{a}_t). \tag{2}$$

The prediction head corresponding to the state token $\mathbf{s}_t$ is trained to predict the associated action $\mathbf{a}_t$. For continuous action spaces, the training objective is to minimize the mean squared error (MSE) loss, defined as:

$$\mathcal{L}_{DT} = \mathbb{E}_{\tau_t \sim \mathcal{D}} \left[ \frac{1}{K} \sum_{i=t-K+1}^{t} (\mathbf{a}_i - \pi(\tau_t)_i)^2 \right], \tag{3}$$

where $\pi(\tau_t)_i$ denotes the $i$-th action output of the policy $\pi$ learned by Equation 3.

### 3.3 Q-LEARNING PENALIZATION

To address the "stitching" challenge and design a target-conditioned policy that aligns the expected returns of sampled actions with the optimal returns while simultaneously minimizing the associated expected cost, we leverage the penalization from the Q-learning module (Kumar et al., 2022; Hu et al., 2024a).

In the safe offline RL setting, two types of Q-networks are utilized: the reward Q-network and the cost Q-network. A straightforward approach to learning these networks involves applying the empirical Bellman evaluation operator, $\mathcal{T}^{\hat{\pi}}$, to samples $(\mathbf{s}, \mathbf{a}, r, c, \mathbf{s}') \sim \mathcal{B}$:

$$Q^r(\mathbf{s}, \mathbf{a}) = r + \gamma \mathbb{E}_{\mathbf{a}' \sim \hat{\pi}(\cdot|\mathbf{s}')} \left[ Q^r(\mathbf{s}', \mathbf{a}') \right], \tag{4}$$

$$Q^c(\mathbf{s}, \mathbf{a}) = c + \gamma \mathbb{E}_{\mathbf{a}' \sim \hat{\pi}(\cdot|\mathbf{s}')} \left[ Q^c(\mathbf{s}', \mathbf{a}') \right], \tag{5}$$

where $Q^r$ and $Q^c$ denote the reward and cost Q-networks, respectively, $\gamma$ represents the discount factor, and $\hat{\pi}$ represents the learned policy by Equation 8.

To mitigate overestimation bias, we employ the double Q-learning technique (Hasselt, 2010), constructing two Q-networks for each type: $Q^r_{\phi_1}, Q^r_{\phi_2}$ for the reward Q-network and $Q^c_{\psi_1}, Q^c_{\psi_2}$ for the cost Q-network. These are accompanied by their corresponding target networks: $Q^r_{\phi'_1}, Q^r_{\phi'_2}, Q^c_{\psi'_1}, Q^c_{\psi'_2}$. Additionally, we construct a target policy $\hat{\pi}_{\theta'}$ to guide the learning process.

Given that the input to the Transformer policy includes trajectory history, we adopt the *n-step Bellman equation* to estimate the Q-networks. This choice is motivated by its demonstrated improvements over the 1-step approximation (Sutton & Barto, 2018). The optimization of the reward Q-network parameters $\phi_i$ for $i \in \{1, 2\}$ is performed by minimizing the following objective:

$$\mathbb{E}_{\tau_t \sim \mathcal{D}, \hat{\mathbf{a}}_t \sim \hat{\pi}_{\theta'}} \sum_{m=t-K+1}^{t-1} \left|\left| \hat{Q}^r_m - Q^r_{\phi_i}(\mathbf{s}_m, \mathbf{a}_m) \right|\right|^2, \tag{6}$$

$$s.t. \ \hat{Q}^r_m = \sum_{j=m}^{t-1} \gamma^{j-m} r_j + \gamma^{t-m} \min_{i=1,2} Q^r_{\phi'_i}(\mathbf{s}_t, \hat{\mathbf{a}}_t),$$

where $\hat{\mathbf{a}}_t$ denotes the predicted action output by the target policy $\hat{\pi}_{\theta'}$. Similarly, the optimization of the cost Q-network parameters $\psi_i$ for $i \in \{1, 2\}$ is performed by minimizing the following objective:

$$\mathbb{E}_{\tau_t \sim \mathcal{D}, \hat{\mathbf{a}}_t \sim \hat{\pi}_{\theta'}} \sum_{m=t-K+1}^{t-1} \left|\left| \hat{Q}^c_m - Q^c_{\psi_i}(\mathbf{s}_m, \mathbf{a}_m) \right|\right|^2, \tag{7}$$

$$s.t. \ \hat{Q}^c_m = \sum_{j=m}^{t-1} \gamma^{j-m} c_j + \gamma^{t-m} \min_{i=1,2} Q^c_{\psi'_i}(\mathbf{s}_t, \hat{\mathbf{a}}_t).$$

Leveraging the learned Q-networks, we incorporate them as penalization mechanisms during the training phase. This approach aims to enhance the policy's "stitching" capability by prioritizing the sampling of high-reward actions while ensuring low-cost trajectories are favored. The final learning objective is formulated as a linear combination of MSE loss and Q-learning penalization terms:

$$\hat{\pi} = \underset{\hat{\pi}_\theta}{\arg\min} \left\{ \mathcal{L}(\theta) := \mathcal{L}_{DT}(\theta) - \mathcal{L}_{Q^r}(\theta) + \mathcal{L}_{Q^c}(\theta) \right\}$$

$$= \underset{\hat{\pi}_\theta}{\arg\min} \underbrace{\mathcal{L}_{DT}(\theta)}_{\text{regularization}} - \alpha_1 \cdot \underbrace{\mathbb{E}_{\tau_t \sim \mathcal{D}} \mathbb{E}_{(\mathbf{s}_i, \mathbf{a}_i) \sim \tau_t} Q_\phi^r(\mathbf{s}_i, \hat{\pi}(\tau_t)_i)}_{\text{reward maximization}} + \alpha_2 \cdot \underbrace{\mathbb{E}_{\tau_t \sim \mathcal{D}} \mathbb{E}_{(\mathbf{s}_i, \mathbf{a}_i) \sim \tau_t} Q_\psi^c(\mathbf{s}_i, \hat{\pi}(\tau_t)_i)}_{\text{cost minimization}}.$$

$$(8)$$

To account for variations in the scale of the Q-networks across different offline datasets, we employ a normalization technique inspired by Fujimoto & Gu (2021). Specifically, the weighting factors $\alpha_1$ and $\alpha_2$ are defined as follows:

$$\alpha_1 = \frac{\eta_1}{\mathbb{E}_{\tau_t \sim \mathcal{D}} \mathbb{E}_{(\mathbf{s}, \mathbf{a}) \sim \tau_t} \left[ \left| Q_\phi^r(\mathbf{s}, \mathbf{a}) \right| \right]}, \tag{9}$$

$$\alpha_2 = \frac{\eta_2}{\mathbb{E}_{\tau_t \sim \mathcal{D}} \mathbb{E}_{(\mathbf{s}, \mathbf{a}) \sim \tau_t} \left[ \left| Q_\psi^c(\mathbf{s}, \mathbf{a}) \right| \right]}, \tag{10}$$

where $\eta_1, \eta_2$ are hyperparameters that control the balance between these loss terms. Notably, these Q-networks in the denominator serve exclusively for normalization and are not subject to differentiation.

## 3.4 DATA AUGMENTATION AND ENSEMBLE

QPT leverages a conditional transformer structure, making the agent's behavior highly sensitive to the selection of target reward and cost values. In the context of safe offline RL, the range of feasible and valid target cost and reward pairs is inherently limited. This limitation poses a significant challenge: how can conflicts between the two target returns be effectively resolved while ensuring that meeting the target cost is prioritized over maximizing the target reward? To address the aforementioned issues, we employ two techniques: data augmentation and ensemble.

Inspired by CDT (Liu et al., 2023b), when an infeasible pair of target reward and cost $(\rho, \kappa)$ arises, we associate the conflicting target with the safest trajectory that achieves the maximum reward:

$$\tau^* = \underset{\tau \sim \mathcal{D}}{\arg\max} \, R(\tau), s.t. \, C(\tau) \leq \kappa. \tag{11}$$

Based on the identified trajectory $\tau^* = \{\hat{r}_t^*, \hat{c}_t^*, \mathbf{s}_t^*, \mathbf{a}_t^*\}_t$, we construct a new augmented trajectory:

$$\hat{\tau} = \{\hat{r}_t^* + \rho - R(\tau^*), \hat{c}_t^* + \kappa - C(\tau^*), \mathbf{s}_t^*, \mathbf{a}_t^*\}_t, \tag{12}$$

where the operation over $\hat{r}^*$ and $\hat{c}^*$ are applied element-wise. This augmentation technique enables the agent to learn by imitating the behavior of the most rewarding and safe trajectory $\tau^*$ when the desired target pair $(\rho, \kappa)$ is infeasible. Further details on this process are provided in Appendix C.2.

Moreover, sequence modeling methods are sensitive to the choice of target conditioning, which serves as input to the policy during inference. Rather than manually tuning the values of the return-to-go and cost-to-go tokens, as required in previous conditional transformer policies – a process that demands extensive trial and error – we leverage learned reward and cost Q-networks to guide action selection. Specifically, actions are preferentially sampled to maximize expected returns while minimizing costs, following the approach in Hu et al. (2024a). This process can be formulated as:

$$\underset{\hat{\mathbf{a}}_t^j}{\arg\max} \quad Q_{\phi'}^r(\mathbf{s}_t, \hat{\mathbf{a}}_t^j), \tag{13}$$

$$s.t. \quad Q_{\psi'}^c(\mathbf{s}_t, \hat{\mathbf{a}}_t^j) \leq \hat{c}_t^j, \tag{14}$$

$$\hat{\mathbf{a}}_t^j = \hat{\pi}(\hat{r}_{t-K+1:t}^j, \hat{c}_{t-K+1:t}^j, \mathbf{s}_{t-K+1:t}, \mathbf{a}_{t-K+1:t-1})).$$

Here, $(\hat{r}^j, \hat{c}^j)$ represent candidate target reward and cost pairs. This approach is highly parallelizable. By assigning distinct return-to-go and cost-to-go pairs to each batch, we can effectively utilize GPU capabilities to concurrently generate multiple action sequences, thereby minimizing computational

overhead. Further details on this process are provided in Appendix C.3 A larger number of candidate target pairs provides a broader search space, potentially improving performance. However, this also incurs increased computational costs and greater susceptibility to noisy or suboptimal pairs, stemming from the biased estimation of the learned Q-networks. Corresponding ablation studies are conducted to demonstrate the efficacy of this procedure, as detailed in Section 4.1 and Appendix D.3. The training and inference procedures are thoroughly outlined in Algorithm 1, providing a comprehensive summary of the processes involved.

### 3.5 THEORETICAL ANALYSIS

In this section, we provide a theoretical analysis of QPT, specifically proving that a safe and high-reward policy can be learned from an offline dataset.

**Theorem 3.1.** *Consider an MDP with binary rewards and costs, behavior policy $\beta$, and conditioning function $f^r$ and $f^c$. Let $g^r(\tau) = \sum_{t=1}^{\mathcal{H}} r_t, g^c(\tau) = \sum_{t=1}^{\mathcal{H}} c_t$. Assume the following:*

1. *Return coverage: $P_\beta(g^r(\tau) = f^r(\mathbf{s}_1)|\mathbf{s}_1) \geq \alpha_{f^r}, P_\beta(g^c(\tau) = f^c(\mathbf{s}_1)|\mathbf{s}_1) \geq \alpha_{f^c}$ for all initial states $\mathbf{s}_1$.*

2. *Near determinism: $P(r \neq \mathcal{R}(\mathbf{s}, \mathbf{a})$ or $c \neq \mathcal{C}(\mathbf{s}, \mathbf{a})$ or $\mathbf{s}' \neq \mathcal{T}(\mathbf{s}, \mathbf{a})|\mathbf{s}, \mathbf{a}) \leq \epsilon$ at all $\mathbf{s}, \mathbf{a}$ for some functions $\mathcal{T}, \mathcal{R}$ and $\mathcal{C}$.*

3. *Consistency of $f^r$ and $f^c$: $f^r(\mathbf{s}) = f^r(\mathbf{s}') + r, f^c(\mathbf{s}) = f^c(\mathbf{s}') + c$ for all $\mathbf{s}$.*

*For timestep $i$, the probabilities of selecting actions with maximum reward or minimum cost satisfy:*

*1. **Reward Selection**: $P\{\hat{P}_i^r - P_i^r \geq \sigma_r, \forall i\} \geq 1 - \delta_r$, where $P_i^r$ and $\hat{P}_i^r$ are probabilities under the policies updated by Equation 3 and Equation 8, respectively. With probability at least $(1 - \delta_r)$:*

$$\mathbb{E}_{\tau \sim \pi^*}[g^r(\tau)] - \mathbb{E}_{\tau \sim \hat{\pi}}[g^r(\tau)] \leq \epsilon(\frac{1}{\alpha_{f^r}} + 3)\mathcal{H}^2 - \mathcal{H}\sigma_r.$$

*2. **Cost Selection**: $P\{\hat{P}_i^c - P_i^c \geq \sigma_c, \forall i\} \geq 1 - \delta_c$, where $P_i^c$ and $\hat{P}_i^c$ are probabilities under the policies updated by Equation 3 and Equation 8, respectively. With probability at least $(1 - \delta_c)$:*

$$\mathbb{E}_{\tau \sim \hat{\pi}}[g^c(\tau)] - \mathbb{E}_{\tau \sim \pi^*}[g^c(\tau)] \leq \epsilon(\frac{1}{\alpha_{f^c}} + 3)\mathcal{H}^2 - \mathcal{H}\sigma_c.$$

Theorem 3.1 demonstrates that training with Equation 8 enables the recovery of near-optimal policies $\pi^*$ from the offline dataset under the specified assumptions, providing the theoretical support for our algorithm. The complete proof and detailed illustration are provided in Appendix B.

## 4 EXPERIMENT

**Experimental Setups.** We conducted extensive evaluations on tasks from *Safety-Gymnasium* (Ray et al., 2019; Ji et al., 2024), *Bullet-Safety-Gym* (Gronauer, 2022), and *MetaDrive* (Li et al., 2022), utilizing the DSRL benchmark (Liu et al., 2023a) to assess the performance of QPT against state-of-the-art safe offline RL methods. The evaluation metrics used are normalized return and normalized cost, where a normalized cost below 1 is indicative of safety. In accordance with the DSRL benchmark, safety is prioritized as the primary evaluation criterion, with higher rewards pursued only after meeting safety requirements. To ensure fair comparisons, we set the cost limit for all tasks to 10.

**Baselines.** We compared our method with four types of baseline methods: (1) Q-learning-based algorithms: CPQ (Xu et al., 2022), BCQ-Lag (Fujimoto et al., 2019; Stooke et al., 2020); (2) Distribution correction estimation: COptiDICE (Lee et al., 2022; 2021); (3) Imitation learning: Behavior Cloning (BC-Safe) (Liu et al., 2023a), which is trained exclusively on safe trajectories that satisfy safety constraints, and FISOR (Zheng et al., 2024), which leverages diffusion models for the development of safe policies; (4) Sequential modeling algorithms: CDT (Liu et al., 2023b), which incorporates cost-to-go token in the training process. The codebase for these baseline methods are sourced from Liu et al. (2023a) and executed by us to ensure a fair comparison. For evaluation, when the normalized cost is below 1, we select the configuration with the highest normalized reward. Otherwise, we prioritize minimizing normalized cost and report the corresponding normalized reward.

Table 1: Complete evaluation results of the normalized reward and cost. The cost threshold is 1. The ↑ symbol denotes that the higher reward, the better. The ↓ symbol denotes that the lower cost (up to threshold 1), the better. Each value is averaged over 20 evaluation episodes and 3 random seeds. **Bold**: Safe agents whose normalized cost is smaller than 1. Gray: Unsafe agents with normalized costs exceeding 1. **Blue**: Safe agent with the highest reward.

| Task | QPT (Ours) | | BC-Safe | | CDT | | BCQ-Lag | | CPQ | | COptiDICE | | FISOR | |
|---|---|---|---|---|---|---|---|---|---|---|---|---|---|---|
| | reward ↑ | cost ↓ | reward ↑ | cost ↓ | reward ↑ | cost ↓ | reward ↑ | cost ↓ | reward ↑ | cost ↓ | reward ↑ | cost ↓ | reward ↑ | cost ↓ |
| PointButton1 | 0.13 | 0.81 | 0.10 | 0.63 | 0.62 | 7.17 | 0.24 | 1.73 | 0.69 | 3.2 | 0.13 | 1.35 | 0.03 | 0.81 |
| PointButton2 | -0.01 | 0.88 | 0.04 | 0.58 | 0.31 | 5.15 | 0.4 | 2.66 | 0.58 | 4.3 | 0.15 | 1.51 | 0.02 | 0.69 |
| PointCircle1 | 0.58 | 0.93 | 0.45 | 0.67 | 0.57 | 0.75 | 0.17 | 1.04 | 0.43 | 0.29 | 0.78 | 15.64 | 0.60 | 12.8 |
| PointCircle2 | 0.62 | 0.92 | 0.49 | 0.44 | 0.61 | 1.39 | 0.53 | 8.35 | 0.28 | 0.77 | 0.78 | 25.94 | 0.70 | 11.79 |
| PointGoal1 | 0.68 | 0.65 | 0.42 | 0.70 | 0.70 | 1.54 | 0.59 | 1.30 | 0.68 | 0.76 | 0.35 | 1.75 | 0.54 | 2.73 |
| PointGoal2 | 0.18 | 0.76 | 0.16 | 0.29 | 0.57 | 3.45 | 0.71 | 7.53 | 0.08 | 2.14 | 0.42 | 2.71 | 0.04 | 0.14 |
| PointPush1 | 0.34 | 0.81 | 0.17 | 0.72 | 0.23 | 1.65 | 0.19 | 1.05 | 0.21 | 0.29 | 0.12 | 0.82 | 0.28 | 0.54 |
| PointPush2 | 0.18 | 0.90 | 0.15 | 0.76 | 0.18 | 1.69 | 0.12 | 1.19 | 0.14 | 0.56 | 0.08 | 1.19 | 0.05 | 0.27 |
| CarButton1 | -0.15 | 0.87 | 0.05 | 0.51 | 0.16 | 4.91 | 0.04 | 1.63 | 0.42 | 9.66 | -0.08 | 1.68 | 0.02 | 0.12 |
| CarButton2 | -0.33 | 0.99 | -0.01 | 0.71 | 0.08 | 5.87 | 0.06 | 2.13 | 0.37 | 12.51 | -0.07 | 1.59 | 0.01 | 0.20 |
| CarCircle1 | 0.31 | 0.39 | 0.21 | 0.95 | 0.45 | 4.62 | 0.59 | 11.06 | -0.09 | 1.02 | 0.64 | 15.47 | 0.60 | 6.54 |
| CarCircle2 | 0.48 | 0.94 | 0.54 | 3.38 | 0.45 | 6.24 | 0.53 | 8.35 | 0.50 | 0.13 | 0.64 | 18.15 | 0.45 | 1.46 |
| CarGoal1 | 0.60 | 0.44 | 0.39 | 0.25 | 0.72 | 2.25 | 0.44 | 2.76 | 0.33 | 4.93 | 0.43 | 2.81 | 0.49 | 0.83 |
| CarGoal2 | 0.28 | 0.64 | 0.19 | 0.68 | 0.39 | 3.53 | 0.34 | 4.72 | 0.10 | 6.31 | 0.19 | 2.83 | 0.06 | 0.33 |
| CarPush1 | 0.34 | 0.65 | 0.23 | 0.35 | 0.34 | 0.79 | 0.23 | 1.33 | 0.08 | 0.77 | 0.21 | 1.28 | 0.28 | 0.28 |
| CarPush2 | 0.18 | 0.61 | 0.10 | 0.91 | 0.11 | 2.33 | 0.10 | 2.78 | -0.03 | 10.00 | 0.10 | 4.55 | 0.14 | 0.89 |
| SwimmerVelocity | 0.65 | 0.58 | 0.55 | 0.89 | 0.65 | 0.94 | 0.29 | 4.10 | 0.31 | 11.58 | 0.58 | 23.64 | -0.04 | 0.00 |
| HopperVelocity | 0.88 | 0.45 | 0.58 | 0.45 | 0.76 | 0.97 | 0.12 | 0.97 | 0.57 | 0.00 | 0.23 | 1.44 | 0.19 | 0.51 |
| HalfCheetahVelocity | 1.01 | 0.03 | 0.90 | 0.53 | 1.01 | 0.28 | 1.04 | 57.06 | 0.08 | 2.56 | 0.43 | 0.00 | 0.89 | 0.00 |
| Walker2dVelocity | 0.83 | 0.47 | 0.81 | 0.31 | 0.83 | 0.97 | 0.81 | 0.37 | 0.31 | 0.65 | 0.09 | 0.84 | 0.23 | 0.83 |
| AntVelocity | 0.99 | 0.78 | 0.96 | 0.89 | 0.98 | 0.94 | 0.85 | 18.54 | -1.01 | 0.00 | 1.00 | 10.29 | 0.89 | 0.00 |
| **SafetyGym Average** | 0.54 | 0.64 | 0.43 | 0.77 | 0.56 | 2.02 | 0.45 | 7.79 | 0.17 | 2.52 | 0.42 | 7.61 | 0.31 | 2.01 |
| BallRun | 0.31 | 0.00 | 0.29 | 0.37 | 0.32 | 1.00 | 0.30 | 0.89 | 0.33 | 0.00 | 0.26 | 0.96 | 0.24 | 0.00 |
| CarRun | 0.99 | 0.30 | 0.98 | 0.34 | 0.99 | 0.78 | 0.98 | 0.13 | 0.98 | 0.23 | 0.95 | 0.54 | 0.76 | 0.00 |
| DroneRun | 0.60 | 0.48 | 0.57 | 0.00 | 0.59 | 0.80 | 0.68 | 4.47 | 0.46 | 0.00 | 0.57 | 6.67 | 0.31 | 0.16 |
| AntRun | 0.73 | 0.82 | 0.70 | 0.79 | 0.72 | 0.99 | 0.58 | 0.77 | 0.09 | 0.46 | 0.61 | 0.92 | 0.52 | 0.83 |
| BallCircle | 0.68 | 0.95 | 0.55 | 0.08 | 0.68 | 0.97 | 0.68 | 1.57 | 0.71 | 0.30 | 0.64 | 3.28 | 0.36 | 0.00 |
| CarCircle | 0.67 | 0.90 | 0.55 | 0.43 | 0.73 | 0.83 | 0.46 | 1.42 | 0.73 | 0.89 | 0.46 | 2.78 | 0.42 | 0.16 |
| DroneCircle | 0.60 | 0.92 | 0.57 | 0.56 | 0.58 | 0.96 | 0.52 | 0.98 | -0.20 | 0.45 | 0.26 | 0.51 | 0.49 | 0.00 |
| AntCircle | 0.41 | 0.41 | 0.45 | 0.98 | 0.31 | 1.25 | 0.57 | 2.11 | 0.02 | 0.00 | 0.10 | 1.31 | 0.29 | 0.00 |
| **BulletGym Average** | 0.62 | 0.60 | 0.58 | 0.44 | 0.61 | 0.95 | 0.60 | 1.54 | 0.39 | 0.29 | 0.48 | 2.12 | 0.42 | 0.14 |
| easysparse | 0.70 | 0.98 | 0.28 | 0.20 | 0.51 | 0.76 | 0.09 | 0.92 | -0.05 | 0.19 | 0.07 | 0.86 | 0.44 | 0.28 |
| eastmean | 0.67 | 0.99 | 0.49 | 0.06 | 0.52 | 0.99 | 0.08 | 0.70 | -0.06 | 0.00 | 0.04 | 0.83 | 0.40 | 0.30 |
| easydense | 0.65 | 0.50 | 0.59 | 0.01 | 0.47 | 0.87 | 0.04 | 0.80 | -0.05 | 0.10 | 0.17 | 1.54 | 0.46 | 0.64 |
| mediumsparse | 0.97 | 0.76 | 0.50 | 0.10 | 0.52 | 0.03 | 0.92 | 0.42 | -0.07 | 0.00 | 0.05 | 0.72 | 0.73 | 0.06 |
| mediummean | 0.97 | 0.66 | 0.36 | 0.05 | 0.68 | 0.97 | 0.03 | 0.68 | -0.06 | 0.00 | 0.09 | 0.77 | 0.52 | 0.01 |
| mediumdense | 0.97 | 0.95 | 0.25 | 0.10 | 0.25 | 0.10 | 0.94 | 0.29 | -0.05 | 0.00 | 0.00 | 0.31 | 0.81 | 0.15 |
| hardsparse | 0.44 | 0.98 | 0.24 | 0.00 | 0.37 | 0.48 | 0.47 | 0.80 | -0.05 | 0.06 | 0.16 | 1.92 | 0.32 | 0.01 |
| hardmean | 0.48 | 0.94 | 0.30 | 0.28 | 0.20 | 0.77 | -0.01 | 0.44 | -0.04 | 0.16 | 0.03 | 0.82 | 0.30 | 0.01 |
| harddense | 0.50 | 0.81 | 0.27 | 0.39 | 0.24 | 0.16 | 0.05 | 0.74 | -0.05 | 0.00 | 0.02 | 0.57 | 0.39 | 0.32 |
| **MetaDrive Average** | 0.71 | 0.84 | 0.36 | 0.13 | 0.42 | 0.57 | 0.29 | 0.64 | -0.05 | 0.10 | 0.07 | 0.93 | 0.49 | 0.20 |

**Metrics.** We use the normalized cost return and the normalized reward return as the evaluation metric for comparison. Denote $r_{max}(\mathcal{M})$ and $r_{min}(\mathcal{M})$ as the maximum empirical reward return and the minimum empirical reward return for task $\mathcal{M}$. The normalized reward is computed by:

$$R_{\text{normalized}} = \frac{R_\pi - r_{min}(\mathcal{M})}{r_{max}(\mathcal{M}) - r_{min}(\mathcal{M})} \times 100, \tag{15}$$

where $R_\pi$ denotes the evaluated reward return of policy $\pi$. While the normalized cost is computed by the ratio between the evaluated cost return $C_\pi$ and the target threshold $\kappa$:

$$C_{\text{normalized}} = \frac{C_\pi + \epsilon}{\kappa + \epsilon}, \tag{16}$$

where $\epsilon$ is a positive number to ensure numerical stability if the threshold $\kappa = 0$. The agent is safe if $C_{\text{normalized}} \leq 1$. Without otherwise statements, we will abbreviate "normalized cost return" as "cost" and "normalized reward return" as "reward" for simplicity.

**Main Results.** The evaluation results are summarized in Table 1. QPT stands out as the only method that consistently achieves satisfactory safety performance across all tasks while also attaining the highest returns in most cases. This highlights its effectiveness in simultaneously ensuring safety and achieving high rewards. In contrast, other methods exhibit significant limitations, either due to severe constraint violations or suboptimal returns. Notably, BC-Safe, which is trained exclusively on safe trajectories, satisfies most safety requirements but demonstrates conservative performance with comparatively lower rewards. Q-learning-based algorithms, including BCQ-Lag and CPQ, as well as the distribution correction estimation-based method, COptiDICE, show inconsistent performance.

Table 2: Impact of different components. Average scores and standard deviations are reported over three random seeds for the *harddense* task in the *MetaDrive* setting and the *HopperVelocity* setting. "Train with $Q^r$" and "Train with $Q^c$" indicate whether the corresponding penalization in Equation 8 is applied. "Data aug." refers to the use of data augmentation, while "Inf. with ensemble" denotes ensemble applied at inference time.

| Exp | Data aug. | Train with $Q^r$ | Train with $Q^c$ | Inf. with ensemble | harddense Reward | harddense Cost | HopperVelocity Reward | HopperVelocity Cost |
|---|---|---|---|---|---|---|---|---|
| 1 | | | | | $0.37 \pm 0.19$ | $1.00 \pm 0.08$ | $0.04 \pm 0.02$ | $1.49 \pm 0.12$ |
| 2 | ✓ | | | | $0.40 \pm 0.05$ | $0.94 \pm 0.04$ | $0.54 \pm 0.03$ | $0.65 \pm 0.03$ |
| 3 | ✓ | ✓ | | | $0.48 \pm 0.08$ | $0.94 \pm 0.05$ | $0.85 \pm 0.05$ | $0.99 \pm 0.04$ |
| 4 | ✓ | | ✓ | | $0.43 \pm 0.06$ | $0.14 \pm 0.10$ | $0.15 \pm 0.02$ | $0.22 \pm 0.01$ |
| 5 | ✓ | ✓ | ✓ | | $0.49 \pm 0.04$ | $0.84 \pm 0.02$ | $0.69 \pm 0.04$ | $0.51 \pm 0.02$ |
| 6 | ✓ | | | ✓ | $0.46 \pm 0.05$ | $0.91 \pm 0.06$ | $0.56 \pm 0.02$ | $0.60 \pm 0.04$ |
| 7 | | ✓ | ✓ | ✓ | $0.50 \pm 0.01$ | $0.93 \pm 0.02$ | $0.66 \pm 0.03$ | $0.56 \pm 0.03$ |
| 8 | ✓ | ✓ | ✓ | ✓ | $0.50 \pm 0.02$ | $0.81 \pm 0.03$ | $0.88 \pm 0.02$ | $0.45 \pm 0.04$ |

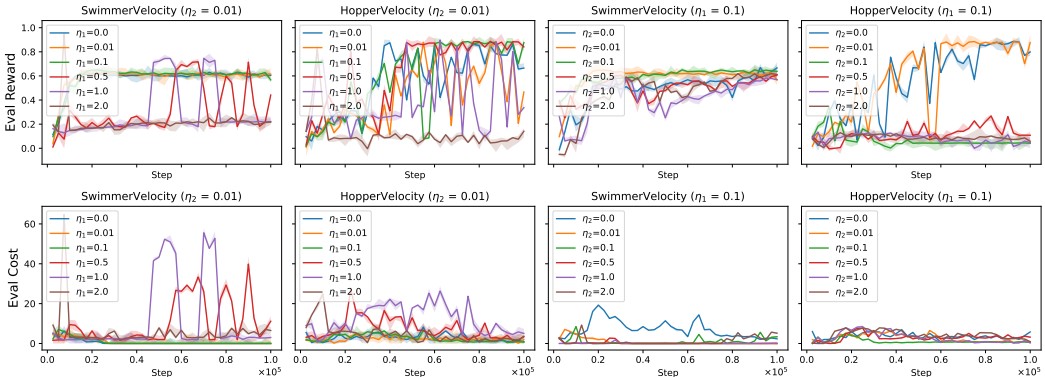

Figure 1: Results of the impact of hyper-parameters $\eta_1$ and $\eta_2$. Each column is a task with one hyper-parameter stable. The x-axis is the training steps. The first row shows the evaluated normalized reward, and the second row shows the evaluated normalized cost. All plots are averaged among 3 random seeds and 20 trajectories for each seed. The solid line is the mean value, and the light shade represents the area within one standard deviation.

These methods tend to oscillate between overly conservative behavior and excessive risk-taking. For instance, CPQ achieves high rewards at the expense of significant safety violations in tasks such as *CarGoal1* and *CarGoal2*, while in *MetaDrive* tasks, it achieves nearly zero cost but at the cost of extremely low rewards. FISOR employs a diffusion-based architecture to maximize rewards within the largest safe region, thereby offering strong safety guarantees. However, it tends to sacrifice potential rewards by strictly adhering to the safe region, resulting in lower reward values while maintaining predominantly safe cost levels. CDT, leveraging its advanced architecture and efficient data utilization, demonstrates more balanced performance. However, it still struggles with trade-offs between safety and utility in safe offline RL settings, particularly in *SafetyGym* tasks, where it fails to meet safety requirements in most cases. In contrast, QPT, which shares the same Transformer architecture as CDT, surpasses it by utilizing our novel framework. QPT achieves the highest returns while consistently satisfying safety requirements, underscoring the efficacy of our proposed method.

## 4.1 ABLATION

**Role of Different Components.** As detailed in Section 3, our methodology incorporates four key components: data augmentation, reward Q-network, cost Q-network, and inference ensemble. Each component warrants individual analysis. We evaluate these components on the *harddense* dataset from the MetaDrive task and on the *HopperVelocity* task, both selected for their challenging nature in achieving high rewards and for the substantial performance improvements that QPT demonstrates over baseline methods. The results are summarized in Table 2. Integrating the $Q^r$ and $Q^c$ networks substantially enhances performance, as evidenced by comparisons between Exp 2 vs. 3 and Exp 2 vs. 4, where the added Q-learning penalization leads to notable improvements in reward and cost metrics. Furthermore, incorporating an ensemble of learned Q-networks further boosts performance, as shown by comparisons between Exp 2 vs. 6 and Exp 5 vs. 8. Data augmentation also improves cost performance by "stitching" additional safe trajectories into the training dataset, as demonstrated

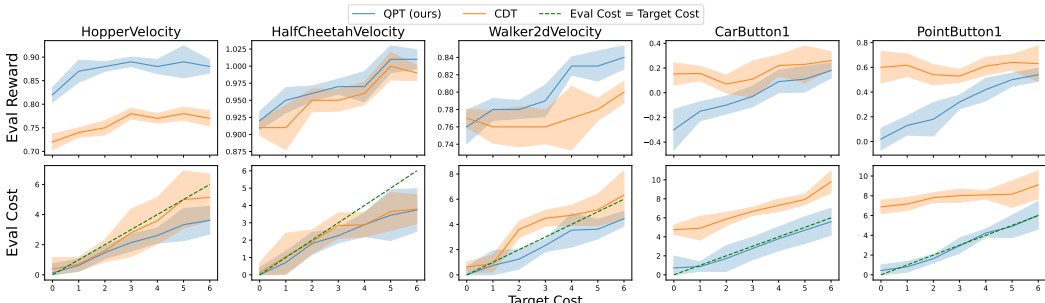

Figure 2: Results of zero-shot adaption to different cost returns. Each column is a task. The x-axis is the target cost return. The first row shows the evaluated normalized reward, and the second row shows the evaluated normalized cost under different target costs. All plots are averaged among 3 random seeds and 20 trajectories for each seed. The solid line is the mean value, and the light shade represents the area within one standard deviation.

by Exp 1 vs. 2 and Exp 7 vs. 8. These findings highlight the effectiveness of our framework in addressing the challenges of safe offline RL.

**Hyper-parameters.** This ablation introduces the hyper-parameters $\eta_1$ and $\eta_2$, as defined in Equation 8, which regulate the influence of two additional loss components. To evaluate their effects, we conducted an ablation study on two tasks: *SwimmerVelocity* and *HopperVelocity*. As illustrated in Figure 1, when $\eta_2$ is held constant and $\eta_1$ is gradually increased, the normalized reward rises within a specific range. For instance, in the *HopperVelocity* task, the reward increases consistently when $\eta_1 \leq 1$. However, beyond this point (e.g., $\eta_1 = 2$), the reward decreases sharply with no further performance gains. Conversely, increasing $\eta_1$ also leads to a corresponding increase in the normalized cost within a certain range, indicating that the policy faces challenges in stitching trajectories with higher associated costs. Similarly, when $\eta_1$ is kept constant and $\eta_2$ is increased, the normalized reward decreases progressively, accompanied by a reduction in the normalized cost within a certain range. This observation suggests that a larger $\eta_2$ can contribute to a safer policy, albeit with a trade-off in reward performance within specific bounds.

**Zero-shot Adaptation.** One significant advantage of the Transformer-based policy is its capability for zero-shot adaptation to varying cost thresholds (Liu et al., 2023b). In contrast, the Q-learning-based baselines introduced earlier lack this capability, as they require a fixed, pre-defined threshold to solve constrained optimization problems. Adapting these methods to new constraint conditions necessitates re-training, which limits their flexibility. Consequently, we primarily compare our method with CDT, which also supports zero-shot adaptation. In this evaluation, each cost-return threshold is treated as a distinct task, with corresponding adjustments made to the target reward. The results are presented in Figure 2. Both methods demonstrate improved performance when conditioned on a higher cost threshold, highlighting the zero-shot adaptation capability of sequence modeling approaches. Furthermore, our method consistently outperforms CDT in scenarios where both methods satisfy the constraint, achieving lower costs than the specified threshold (as shown in the left three plots of Figure 2). Although CDT achieves higher rewards than our method in the Button environments, it does so at the expense of greater safety violations. In contrast, our method adheres to the cost threshold, underscoring its effectiveness in maintaining safety while delivering competitive performance.

## 5 CONCLUSION

This paper addresses the safe offline RL problem from the perspective of trilogy optimization, introducing the Q-learning Penalized Transformer policy (QPT) framework. By integrating Q-learning penalization into the conditional transformer policy, QPT effectively maximizes expected rewards while minimizing expected costs. Theoretical analysis under mild assumptions highlights its advantages in aligning with optimal policies. Extensive experiments demonstrate QPT's ability to learn safe, robust, high-reward policies, consistently outperforming state-of-the-art baselines and retaining zero-shot adaptation to varying constraints. We hope this work inspires further research into safety and generalization in offline learning.

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

## A    ADDITIONAL STATEMENT

**The Use of Large Language Models.**    In this work, we exclusively employ large language models (LLMs) to refine the writing and presentation of our manuscript.

**Limitations.**    The theoretical guarantees of our algorithms rely on a near-deterministic environment, an assumption that does not always hold in real-world deep learning models. Moreover, current evaluations rely on limited offline data, potentially constraining performance. Extending QPT to a safe offline-to-online RL framework presents a promising direction to enhance performance through online interactions while maintaining safety.

## B    PROOF OF THEOREM 3.1

First, we give the following Lemma.

**Lemma B.1** (Alignment with respect to the conditioning function (Brandfonbrener et al., 2022))**.** *Consider an MDP, behavior $\beta$ and conditioning function $f^r$. Let $J^r(\pi) = \mathbb{E}_{\tau \sim \pi}[g^r(\tau)]$, where $g^r(\tau) = \sum_{t=1}^{\mathcal{H}} r_t$. Assume the following:*

    *1. Return coverage: $P_\beta(g^r(\tau) = f^r(\mathbf{s}_1)|\mathbf{s}_1) \geq \alpha_{f^r}$ for all initial states $\mathbf{s}_1$.*

    *2. Near determinism: $P(r \neq \mathcal{R}(\mathbf{s}, \mathbf{a})$ or $s' \neq \mathcal{T}(\mathbf{s}, \mathbf{a})|\mathbf{s}, \mathbf{a}) \leq \epsilon$ at all $s, a$ for some functions $\mathcal{T}$ and $\mathcal{R}$. Note that this does not constrain the stochasticity of the initial state.*

    *3. Consistency of $f^r$: $f^r(\mathbf{s}) = f^r(\mathbf{s}') + r$ for all $\mathbf{s}$.* [1]

*Then*

$$J^r(\pi^*) - J^r(\pi) \leq \epsilon \left( \frac{1}{\alpha_{f^r}} + 3 \right) \mathcal{H}^2, \tag{17}$$

*where $\pi$ is derived from Equation 3, , $\pi^*$ is the optimal policy, and $\mathcal{H}$ is the horizon length of episode. Moreover, there exist problems where the bound is tight up to constant factors.*

**Corollary B.2.** *Under assumptions analogous to those in Lemma B.1 for the cost function, specifically: $P_\beta(g^c(\tau) = f^c(\mathbf{s}_1)|\mathbf{s}_1) \geq \alpha_{f^c}$, $P(c \neq \mathcal{C}(\mathbf{s}, \mathbf{a})$ or $s' \neq \mathcal{T}(\mathbf{s}, \mathbf{a})|\mathbf{s}, \mathbf{a}) \leq \epsilon$, $f^c(\mathbf{s}) = f^c(\mathbf{s}') + c$. Let $J^c(\pi) = \mathbb{E}_{\tau \sim \pi}[g^c(\tau)]$, the following bound holds:*

$$J^c(\pi) - J^c(\pi^*) \geq \epsilon \left( \frac{1}{\alpha_{f^c}} + 3 \right) \mathcal{H}^2. \tag{18}$$

The proof follows a similar reasoning as Lemma B.1, given the structural parallels between the cost and reward functions.

Based on Lemma B.1 and Corollary B.2, we now give the proof of Theorem 3.1.

*Proof.* We prove the bounds for reward and cost separately.

**Reward Bound.** For the reward, we begin with:

$$\mathbb{E}_{\tau \sim \pi^*}[g^r(\tau)] - \mathbb{E}_{\tau \sim \hat{\pi}}[g^r(\tau)] \tag{19}$$

$$= \mathbb{E}_{\tau \sim \pi^*}[g^r(\tau)] - \mathbb{E}_{\tau \sim \pi}[g^r(\tau)] + \mathbb{E}_{\tau \sim \pi}[g^r(\tau)] - \mathbb{E}_{\tau \sim \hat{\pi}}[g^r(\tau)] \tag{20}$$

$$= J^r(\pi^*) - J^r(\pi) + J^r(\pi) - \mathbb{E}_{\tau \sim \hat{\pi}}[g^r(\tau)] \tag{21}$$

$$\leq \epsilon \left( \frac{1}{\alpha_f} + 3 \right) \mathcal{H}^2 + J^r(\pi) - \mathbb{E}_{\tau \sim \hat{\pi}}[g^r(\tau)]. \tag{22}$$

---

[1]Note this can be exactly enforced (as in prior work) by augmenting the state space to include the cumulative reward observed so far.

Next, for the second term in Equation 22:

$$J^r(\pi) - \mathbb{E}_{\tau \sim \hat{\pi}}[g^r(\tau)] \tag{23}$$

$$= \mathbb{E}_{\tau \sim \pi}[g^r(\tau)] - \mathbb{E}_{\tau \sim \hat{\pi}}[g^r(\tau)] \tag{24}$$

$$= \mathbb{E}_{\tau \sim \pi}[\sum_{t=1}^{\mathcal{H}}(r_t)] - \mathbb{E}_{\tau \sim \hat{\pi}}[\sum_{t=1}^{\mathcal{H}}(r_t)] \tag{25}$$

$$= \mathbb{E}_{\mathbf{s}_1} \sum_{t=1}^{\mathcal{H}}(P_t^r \cdot r_t) - \mathbb{E}_{\mathbf{s}_1} \sum_{t=1}^{\mathcal{H}}(\hat{P}_t^r \cdot r_t), \tag{26}$$

$$\tag{27}$$

where $P_t^r$ and $\hat{P}_t^r$ represent the probabilities of selecting the maximum-reward actions under policies derived from Equation 3 and Equation 8, respectively. Since rewards are binary and by the condition $P\{\hat{P}_i^r - P_i^r \geq \sigma_r, \forall i\} \geq 1 - \delta_r$, we have:

$$\mathbb{E}_{\mathbf{s}_1} \sum_{t=1}^{\mathcal{H}}(P_t^r \cdot r_t) - \mathbb{E}_{\mathbf{s}_1} \sum_{t=1}^{\mathcal{H}}(\hat{P}_t^r \cdot r_t) \tag{28}$$

$$= \mathbb{E}_{\mathbf{s}_1} \sum_{t=1}^{\mathcal{H}}[(P_t^r - \hat{P}_t^r)r_t] \tag{29}$$

$$\leq \mathbb{E}_{\mathbf{s}_1} \sum_{t=1}^{\mathcal{H}}(-\sigma_r) \cdot r_t \tag{30}$$

$$\leq -\mathcal{H}\sigma_r. \tag{31}$$

Substituting Equation 31 into Equation 22, we get:

$$\mathbb{E}_{\tau \sim \pi^*}[g^r(\tau)] - \mathbb{E}_{\tau \sim \hat{\pi}}[g^r(\tau)] \tag{32}$$

$$\leq \epsilon \left(\frac{1}{\alpha_f} + 3\right) \mathcal{H}^2 - \mathcal{H}\sigma_r. \tag{33}$$

**Cost Bound.** For the cost bound, using a similar approach:

$$\mathbb{E}_{\tau \sim \hat{\pi}}[g^c(\tau)] - \mathbb{E}_{\tau \sim \pi^*}[g^c(\tau)] \tag{34}$$

$$\geq \epsilon \left(\frac{1}{\alpha_f} + 3\right) \mathcal{H}^2 + \mathbb{E}_{\tau \sim \hat{\pi}}[g^c(\tau)] - J^c(\pi) \tag{35}$$

$$= \epsilon \left(\frac{1}{\alpha_f} + 3\right) \mathcal{H}^2 + \mathbb{E}_{\mathbf{s}_1} \sum_{t=1}^{\mathcal{H}}(\hat{P}_t^c \cdot c_t) - \mathbb{E}_{\mathbf{s}_1} \sum_{t=1}^{\mathcal{H}}(P_t^c \cdot c_t), \tag{36}$$

$$= \epsilon \left(\frac{1}{\alpha_f} + 3\right) \mathcal{H}^2 + E_{\mathbf{s}_1} \sum_{t=1}^{\mathcal{H}}[(\hat{P}_t^c - P_t^c)c_t] \tag{37}$$

$$\geq \epsilon \left(\frac{1}{\alpha_f} + 3\right) \mathcal{H}^2 - \mathcal{H}\sigma_c, \tag{38}$$

where $P_t^c$ and $\hat{P}_t^c$ represent the probabilities of selecting minimum-cost actions under policies derived from Equation 3 and Equation 8, respectively. □

*Remark* B.3. We impose three assumptions. *(i)* The offline dataset provides sufficient coverage of the relevant support of returns and costs - i.e., it spans heterogeneous return and cost distributions adequate for conditioning via $f^r$ and $f^c$. *(ii)* The environment dynamics are deterministic or nearly so. *(iii)* The conditioning functions are time-consistent, coinciding with the notions of return-to-go and cost-to-go in our formulation. Under these conditions, the theorem guarantees that, in (near-)deterministic settings with appropriately specified conditioning and adequate data coverage, our method can recover a policy whose performance approaches optimality with probability at least $(1 - \delta_r)$ for reward and $(1 - \delta_c)$ for cost.

*Remark* B.4. The additional Q-learning penalization encourages the learned policy to prioritize higher-reward actions with lower costs. However, due to the constraint feasible region imposed by the two Q-networks, the policy may not always successfully select the desired action. To address this limitation, we introduce additional assumptions regarding reward and cost action selection, ensuring a high probability of selecting the appropriate actions. These assumptions reinforce the effectiveness and reliability of the proposed approach.

*Remark* B.5. Since $J^r(\pi^*)$ represents the expected maximum reward of the optimal policy and $J^c(\pi^*)$ represents the expected minimum cost, the derived bounds for reward and cost naturally align in opposite directions, reflecting their inherently inverse relationship. Compared to Lemma B.1 and Corollary B.2, our framework improves upon the original loss in Equation 3 by an additional $\mathcal{H}\sigma_r$ and $\mathcal{H}\sigma_c$, demonstrating its effectiveness in achieving superior policies compared to Transformer-based baselines.

**Corollary B.6.** *If $\alpha_{f^r} > 0, \epsilon = 0$, and $f^r(s_1) = V^{r*}(s_1)$ for all initial states $s_1$, then $J^r(\pi^*) = J^r(\pi) = J^r(\hat{\pi})$ under $\hat{P}_i^r = P_i^r$ and $\sigma_r = 0$. Analogously, if $\alpha_{f^c} > 0, \epsilon = 0$, and $f^c(s_1) = V^{c*}(s_1)$ for all $s_1$, then $J^c(\pi^*) = J^c(\pi) = J^c(\hat{\pi})$.*

*Remark* B.7. In general, the reward- and cost-side conditions cannot be satisfied simultaneously. The joint feasible set induced by the two $Q$-networks typically imposes conflicting constraints, making $f^r(s_1) = V^{r*}(s_1)$ and $f^c(s_1) = V^{c*}(s_1)$ hold at the same time only in exceptional (e.g., degenerate or perfectly aligned) environments. Hence, it is usually unrealistic to expect both equalities to be achieved concurrently.

## C  ALGORITHM DETAILS

### C.1  ALGORITHM PSEUDOCODE

The detailed pipeline of QPT is summarized in Algorithm 1.

### C.2  DATA AUGMENTATION

The intuition is to relabel the associated Pareto trajectory's reward and cost returns, such that the agent can learn to imitate the behavior of the most rewarding and safe trajectory $\tau^*$ when the desired return $(\rho, \kappa)$ is infeasible, i.e., $\rho > \text{RF}(\kappa, \mathcal{D})$. The Reward Frontier (RF) value is defined by the maximum reward with cost $\kappa \in \mathbb{C}$, where $\mathbb{C} := \{C(\tau) : \tau \in \mathcal{D}\}$ is the set of all the possible episodic cost in $\mathcal{D}$:

$$\text{RF}(\kappa, \mathcal{D}) = \max_{\tau \in \mathcal{D}} R(\tau), \ \ s.t. \ \ C(\tau) = \kappa. \tag{39}$$

The augmentation procedure is detailed in Algorithm 2 (Liu et al., 2023b). Figure 3 provides an illustrative example: arrows map Pareto-optimal trajectories to their corresponding augmented return-cost pairs.

### C.3  ENSEMBLE

In this section, we highlight the detailed ensemble process. During the training phase, RTG and CTG values are derived directly from the trajectory data within the dataset. Specifically, these values are computed as the cumulative discounted rewards and costs from each state to the terminal state along the observed trajectories, thereby preserving the ground-truth signal from the environment. For inference, we utilize the default RTG and CTG pairs established in the DSRL benchmark as our baseline. To generate candidate pairs, we perturb the RTG values by introducing random noises while maintaining constant CTG values across all candidates. This asymmetric perturbation strategy is theoretically motivated: our objective is to maximize expected returns while adhering to a fixed cost constraint. By holding CTG constant while exploring a diverse range of RTG values, we effectively search the action-value landscape for optimal policies that maximize reward within the predetermined cost threshold.

The ensemble inference mechanism represents a computationally efficient approach to action selection that leverages our learned Q-networks to identify optimal actions from multiple candidates. This process operates exclusively during the inference phase and involves the following structured procedure:

---

**Algorithm 1** QPT: Q-learning Penalized Transformer

---

**Input:** Sequence horizon $K$, offline datasets $\mathcal{D}$, coefficient $\rho$, a set of candidate pairs of return-to-go and cost-to-go $\{(\hat{r}_0^0, \hat{c}_0^0), (\hat{r}_0^1, \hat{c}_0^1), \ldots, (\hat{r}_0^m, \hat{c}_0^m)\}$.

Initialize policy network $\pi_\theta$, reward Q-networks $Q_{\phi_1}^r, Q_{\phi_2}^r$, cost Q-networks $Q_{\psi_1}^c, Q_{\psi_2}^c$, and target networks $\pi_{\theta'}, Q_{\phi_1'}^r, Q_{\phi_2'}^r, Q_{\psi_1'}^c$ and $Q_{\psi_2'}^c$.

Update the dataset $\mathcal{D}$ with data augmentation technique based on Equation 12.

// Train the QPT

**for** $t = 1$ **to** $\mathcal{H}$ **do**

    Sample sequence transition mini-batch $\mathcal{B} = \{(\hat{r}_j, \hat{c}_j, \mathbf{s}_j, \mathbf{a}_j, r_j, c_j)_{j=t}^{t+K}, \} \sim \mathcal{D}$.

    // Reward Q-network and cost Q-network learning

    Sample $\hat{\mathbf{a}}_{t+K} \sim \pi_{\theta'}(\hat{\mathbf{a}}_{t+K}|\hat{r}_{t:t+K}, \hat{c}_{t:t+K}, \mathbf{s}_{t:t+K}, \mathbf{a}_{t:t+K-1})$.

    Update $Q_{\phi_1}^r$ and $Q_{\phi_2}^r$ by Equation 6, update $Q_{\psi_1}^c$ and $Q_{\psi_2}^c$ by Equation 7.

    // Policy learning

    **for** $i = 1$ **to** $K$ **do**

        Sample $\hat{\mathbf{a}}_{t+i} \sim \pi_\theta(\hat{\mathbf{a}}_{t+i}|\hat{r}_{t:t+i}, \hat{c}_{t:t+i}, \mathbf{s}_{t:t+i}, \mathbf{a}_{t:t+i-1})$ in an auto-regressive way.

    **end for**.

    Update policy by minimizing Equation 8.

    $\theta' = \rho\theta' + (1-\rho)\theta, \phi_i' = \rho\phi_i' + (1-\rho)\phi_i, \psi_i' = \rho\psi_i' + (1-\rho)\psi_i$ for $i = \{1, 2\}$.

**end for**.

// Inference with QPT

Given multiple pairs of target return-to-go and target cost-to-go choice $(\hat{r}^j, \hat{c}^j)_0^{j=1:m}$ and initial state $s_0$.

**repeat**

    Sample multiple actions with different return-to-go $\hat{\mathbf{a}}_t^j = \pi_\theta(\hat{\mathbf{a}}_t^j|\hat{r}_{t-K+1:t}^j, \hat{c}_{t-K+1:t}^j, \mathbf{s}_{t-K+1:t}, \mathbf{a}_{t-K+1:t-1})$ for $j = 1, \ldots, m$.

    Compute Q networks with candidate state-action pair $(\mathbf{s}_t, \hat{\mathbf{a}}_t^j)$ for $j = 1, \ldots, m$.

    Sample the action $\mathbf{a}_t$ from action set $\{\hat{\mathbf{a}}_t^j\}_{j=1}^m$ with Equation 13 and Equation 14.

    Execute the action $\mathbf{a}_t$ and collect the reward $r_t$, cost $c_t$ and next state $\mathbf{s}_{t+1}$.

    Update current return-to-go $\hat{r}_{t+1}^j = \hat{r}_t^j - r_t$, cost-to-go $\hat{c}_{t+1}^j = \hat{c}_t^j - c_t$ for $j = 1, \ldots, m$.

**until** $Done$ is $true$.

---

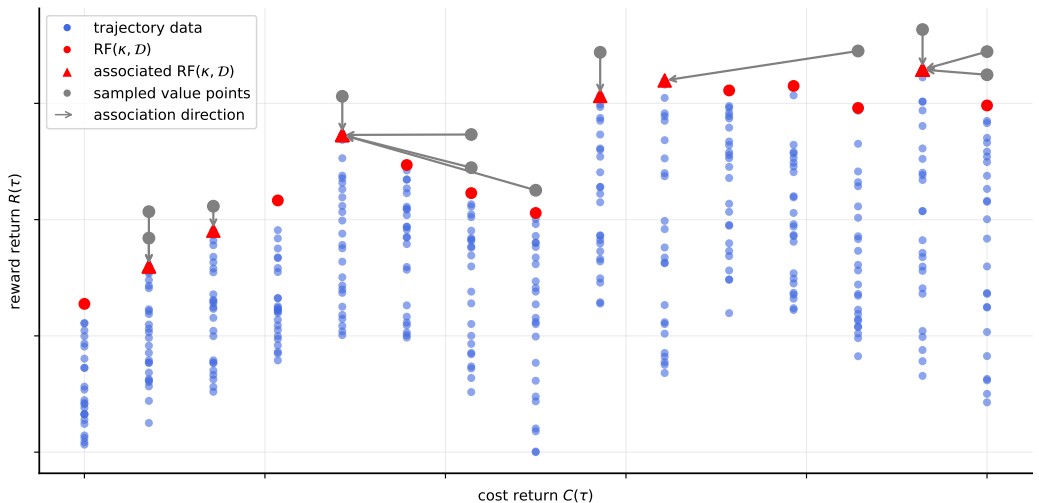

Figure 3: Illustrative example of the data-augmentation procedure (Liu et al., 2023b).

First, we generate multiple return-to-go and cost-to-go conditioning pairs by introducing controlled stochastic perturbations to a default reference pair. This creates a diverse set of conditioning signals that explore different regions of the reward-safety trade-off space. Rather than processing these pairs sequentially, we exploit the parallelization capabilities of modern GPU architectures by batching all candidate pairs into a single forward pass through our trained Transformer model. This parallelization

---

**Algorithm 2** Data Augmentation via Relabeling

---

**Input:** dataset $\mathcal{D}$, samples $N$, reward sample max $r_{max}$
**Output:** augmented trajectory dataset $\mathcal{D}$

1: $c_{min} \leftarrow \min_{\tau \sim \mathcal{D}} C(\tau)$, $c_{max} \leftarrow \max_{\tau \sim \mathcal{D}} C(\tau)$
2: **for** $i = 1, ..., N$ **do**
3:     ▷ *sample a cost return*
4:     $\kappa_i \sim \text{Uniform}(c_{min}, c_{max})$
5:     ▷ *sample a reward return above the RF value*
6:     $\rho_i \sim \text{Uniform}(\text{RF}(\kappa_i, \mathcal{D}), r_{max})$
7:     ▷ *find the closest and safe Pareto trajectory*
8:     $\tau_i^* \leftarrow \arg\max_{\tau \sim \mathcal{D}} R(\tau), s.t. \quad C(\tau) \leq \kappa_i$
9:     ▷ *relabel the reward and cost return*
10:    $\hat{\tau}_i \leftarrow \{\hat{r}_i^* + \rho_i - R(\tau_i^*), \hat{c}_i^* + \kappa_i - C(\tau_i^*), \mathbf{s}_i^*, \mathbf{a}_i^*\}$
11:    ▷ *append the trajectory to the dataset*
12:    $\mathcal{D} \leftarrow \mathcal{D} \cup \{\hat{\tau}_i\}$
13: **end for**

---

technique ensures that the computational overhead remains minimal compared to evaluating a single conditioning pair. Once the model generates actions corresponding to each conditioning pair, we employ our learned Q-networks ($Q^r$ and $Q^c$) as evaluation metrics to select the optimal action according to specified criteria. Depending on the deployment context, these criteria may prioritize reward maximization subject to hard safety constraints, or implement a parameterized trade-off between reward and safety considerations. By dynamically evaluating multiple conditioning pairs during inference, our method effectively automates this hyperparameter selection process, reducing the need for exhaustive offline tuning while potentially discovering superior action candidates that might be overlooked in a single-sample approach.

## D EXPERIMENT DETAILS

### D.1 ENVIRONMENT DESCRIPTIONS

The environments designed for evaluating safe offline RL methods are based on different simulators, each tailored to specific tasks and agent types. Figure 4 visualize some representative tasks of these environments.

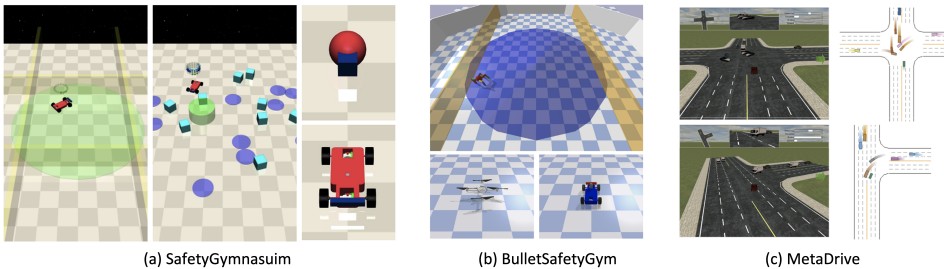

(a) SafetyGymnasuim      (b) BulletSafetyGym      (c) MetaDrive

Figure 4: Visualization of the simulation environments and representative tasks (Liu et al., 2023a).

**Safety-Gymnasium** (Ray et al., 2019; Ji et al., 2024): Built on the Mujoco physics simulator, Safety-Gymnasium provides safety-critical environments with diverse tasks. The Car agent engages in tasks such as Button, Push, and Goal, each available in two difficulty levels. These tasks require navigating hazards while completing objectives. For example, in Goal, the agent moves toward randomly reset goal positions upon completion. In Push, it moves a box to dynamic goal locations, while Button involves pressing scattered goal buttons. Additional velocity-constrained tasks are included for agents such as Ant, HalfCheetah, and Swimmer. The Velocity task challenges agents to coordinate leg movements to move forward, while Run requires navigating from a random direction and speed to a designated endpoint. The Circle task rewards agents for following a circular path while avoiding

hazardous zones. Tasks are named by combining agent, task, and difficulty level (e.g., CarPush1), reflecting complexity and objectives.

**Bullet-Safety-Gym** (Gronauer, 2022): Developed with the PyBullet physics simulator, this suite includes four agent types–Ball, Car, Drone, and Ant–and two primary tasks: Circle and Run. In Run, agents navigate corridors bounded by safety lines, incurring penalties for crossing them or exceeding speed limits. In Circle, agents move clockwise along a circular path, earning rewards for higher speeds near the boundary and penalties for straying outside the safety zone. These environments focus on safety evaluation with shorter, more straightforward tasks compared to Safety-Gymnasium.

**MetaDrive** (Li et al., 2022): MetaDrive is a self-driving simulation environment based on the Panda3D game engine, offering realistic driving conditions with varying road complexity (easy, medium, hard) and traffic density (sparse, medium, dense). Tasks are named by their road and vehicle conditions. This environment enables testing offline RL algorithms in scenarios that closely mimic real-world driving challenges.

An overview of these environments and tasks is presented in Table 3. Each environment presents unique challenges for safe offline RL evaluation, from self-driving simulations to hazard-avoidance tasks, offering varied complexities and objectives for testing algorithm robustness.

Table 3: Overview of the safe RL benchmarks and tasks for dataset collection (Liu et al., 2023a).

| Benchmarks | Backends | Environments | Agents | Difficulty Levels | Total Tasks | Dataset Trajectories |
|---|---|---|---|---|---|---|
| SafetyGymnasium | Mujoco | Goal, Button, Push, Circle | Point, Car | 2 | 16 | 40310 |
| | | Velocity | Ant, HalfCheetah, Hopper, Swimmer, Walker2d | 1 | 5 | 11399 |
| BulletSafetyGym | PyBullet | Run, Circle | Ball, Car, Drone, Ant | 1 | 8 | 14498 |
| MetaDrive | Panda3D | Driving | Vehicle | 3 | 9 | 9000 |

## D.2 HYPERPARAMETERS

The reward and cost Q-networks used across all tasks consist of four linear layers, each employing Mish activation functions for non-linearity. To ensure a fair comparison between QPT and the baseline methods, we use a consistent setup of $10^5$ gradient steps (except for the *MetaDrive* tasks) and a rollout length equal to the maximum episode length for all experiments. A comprehensive list of hyperparameters utilized in the experiments is provided in Table 4.

Table 4: Hyperparameters for QPT

| Parameter | All tasks | Parameter | All tasks |
|---|---|---|---|
| Number of layers | 3 | Number of attention heads | 8 |
| Embedding dimension | 128 | Batch size | 2048 |
| Context length $K$ | 10 | Learning rate | 0.0001 |
| Droupout | 0.1 | Adam betas | (0.9, 0.999) |
| Grad norm clip | 0.25 | Cost threshold | 10 |
| Training steps (BulletGym, SafetyGym) | 100000 | Training steps (MetaDrive) | 200000 |

## D.3 ABLATION OF THE NUMBER OF CANDIDATE ACTIONS IN ENSEMBLE

In our implementation, we utilize a default ensemble size of 50 candidate actions during inference. We also conduct a systematic ablation study specifically examining the impact of ensemble size on performance using the MetaDrive harddense environment as our testbed. The results in Table 5 reveal a nuanced relationship between ensemble size and overall performance. As the number of sampled candidates increases from small values, we observe consistent performance improvements, indicating that larger ensemble sizes enable more comprehensive exploration of the action space and higher-quality policy selection. However, this relationship exhibits clear non-monotonicity, with

performance plateauing and eventually declining beyond a certain threshold. A larger number of candidate target pairs provides a broader search space, potentially improving performance. However, this also incurs increased computational costs and greater susceptibility to noisy or suboptimal pairs, stemming from the biased estimation of the learned Q-networks.

Table 5: Impact of the number of candidate target reward and cost pairs in the *harddense* task in the *MetaDrive* setting.

|  | 1 | 10 | 30 | 50 | 100 |
|---|---|---|---|---|---|
| Reward | $0.49 \pm 0.04$ | $0.50 \pm 0.02$ | $0.52 \pm 0.03$ | $0.50 \pm 0.02$ | $0.48 \pm 0.04$ |
| Cost | $0.84 \pm 0.02$ | $0.83 \pm 0.03$ | $0.90 \pm 0.05$ | $0.81 \pm 0.03$ | $0.80 \pm 0.03$ |

