# OpenReview forum: "Q-learning Penalized Transformer for Safe Offline Reinforcement Learning"
_ICLR.cc/2026/Conference — Submitted to ICLR 2026_

### Official Review · Reviewer_BCLB · 2025-10-30

**Soundness:** 2
**Presentation:** 3
**Contribution:** 2
**Rating:** 4
**Confidence:** 4

**Summary:**

This paper proposes QPT (Q-Learning Penalized Transformer), a safe offline RL method that integrates Q-learning with a Transformer-based policy to jointly optimize reward, cost, and behavior regularization. It guarantees near-optimal safe policies and demonstrates better safety and performance across DSRL benchmark tasks.

**Strengths:**

The paper provides theoretical results claiming that the proposed method can learn policies that are both safe and high-reward.

**Weaknesses:**

The proposed method and theoretical analysis mainly restate existing intuitions without offering substantial novelty. For instance, Eq. (8) closely resembles the policy loss in TD3+BC, combining an MSE action loss with $Q$-function regularization. The formulation for $Q_c$​ is a direct analogy of $Q_r$​, such as Eq.(10), and the main theorem extends RCSL in a straightforward manner by adding a parallel analysis for the cost function, with the proofs largely following the same structure as those in RCSL.

**Questions:**

1. The $Q$-functions are trained using the n-step Bellman equation. How does the proposed method handle the out-of-distribution (OOD) action issue, which is a well-known challenge in offline RL? Without explicit regularization or behavior constraints, it is unclear how stability is maintained when extrapolating beyond the dataset’s support.

2. The data augmentation technique appears to be directly adopted from CDT. What is the motivation for reintroducing this component in Section 3.4 and performing an ablation study on it, given that its empirical benefits have already been established in the CDT paper?

3. Regarding the main results and zero-shot adaptation to varying cost thresholds, several key baselines are missing, including TREBI (already cited in the references), CAPS [1], and CCAC [2]. TREBI leverages diffusion models for zero-shot generalization, while CAPS and CCAC are value-based methods that also achieve zero-shot generalization. The absence of these comparisons makes it difficult to fully assess the claimed effectiveness of the proposed approach, especially since CDT has already demonstrated that transformer-based policies conditioned on cost can achieve similar generalization.

4. As shown in Table 2, the variant trained without $Q_c$​ still maintains safety performance. Does this imply that the primary factor contributing to safety satisfaction comes from the transformer-based policy architecture rather than the proposed cost-related $Q_c$ components? This observation seems to weaken the claimed contributions; could the authors clarify this point?

[1] Chemingui et al., Constraint-Adaptive Policy Switching for Offline Safe Reinforcement Learning, AAAI 2025. \
[2] Guo et al., Constraint-Conditioned Actor-Critic for Offline Safe Reinforcement Learning, ICLR 2025.

---

### Official Review · Reviewer_7Y5D · 2025-10-30

**Soundness:** 2
**Presentation:** 1
**Contribution:** 2
**Rating:** 2
**Confidence:** 5

**Summary:**

This paper proposes an safe offline RL algorithm by extending the Constrained Decision Transformer framework.
The paper's main contributions are:
1. Two Q-functions are introduced to improve the ability of maximizing reward and minimizing costs by adding penalization to CDT's loss.
2. The Q-functions are also utilized in ensemble during inference time to filter safe actions with high value.

The proposed method is evaluated on 38 tasks, demonstrating improved safety over baseline methods.

**Strengths:**

1. The paper introduces Q-learning as penalization term adding to the loss of CDT to address the stitching challenge.

2. QPT can adapt to varying constraints without retraining.

3. QPT proposes inference-time ensemble to further improve of maximizing reward and minimizing constraint violation.

4. This paper tests QPT on enough benchmarks to demonstrate the performance.

**Weaknesses:**

## The writing logic of this paper is confusing, including the central claims, methodology, and related work.

### 1. Central claims

1. The challenging problem in SORL requires balancing three highly interconnected and competing objectives: (1) constraint satisfaction, (2) reward maximization, and (3) OOD problem imposed by offline dataset. This is the indisputable consensus of all safe offline RL articles.
2. This paper replaces the third problem to “behavior regularization”. Although behavior regularization is one of methods to mitigate OOD problem, it is not accurate representation here since other methods can also address the OOD problem, such as trajectory augmentation or generation (TREBI [1], OASIS [2]).
   1. Further explanation are shown in **3. Reward work** part.

3. Besides, the detailed claim of this paper is not clear (At least, it hasn't been clearly written). The above challenge is the largest and most general challenge for all Safe Offline RL algorithms. This challenge cannot reflect the specific contribution of QPT different from other methods.

### 2. Method

This paper is an incremental paper of CDT (Section 3.2 and part of 3.4). This paper mixed **preliminary** and **methodology** section together to one **methodology** section. The contribution of this paper is limited after excluding the **preliminary** content:

1. Section 3.1 should belong to **preliminary** section since it is problem setup rather than proposed by this paper.
2. Section 3.2 also belongs to **preliminary** section, since it is a description of CDT without any improvement.
3. Section 3.3 is the main contribution of this paper while the core idea is similar to ADT, which improve stitching ability by Q-network in Offline RL.
4. Section 3.4 partially belongs to CDT and partially belongs to the contribution.
   1. **What is your improvement on data augmentation compared with CDT.**
   1. **Algorithm 2 in this paper seems to be totally same as the Algorithm 1 in CDT.**
   1. **At least, the differences are not explained clearly**

Please move all parts of previous work to a independent **preliminary** section so that reviewer can clearly know your contribution.

Questions:

1. For Q-networks
   1. In online setting, the n-step rewards/costs come from the current policy via sampling online to make sure all values are from the same policy. However, the n-step rewards/costs come from the policy to collect the dataset (behavior policy) rather than the current policy in this paper. How to explain this issue? Is there any influence? Why is 1-step not suitable in this paper?
   2. CDT avoids the OOD action problem since it does not utilize any Bellman backup procedure. However, QPT trains the two Q-networks with Bellman backup procedure. How to address the corresponding OOD  problem since it is one of the challenging problem in central claims?

### 3. Related work

1. In offline RL, the major challenge is not distribution shift. RL is an optimization method. To be optimized, RL must shift the distribution from behavior policy to optimal policy. The reason why offline RL needs to constrain the target policy near the behavior policy is **to mitigate the OOD problem**. This problem happens during the bellman backup procedure when bootstrapping. The target Q function may wrongly estimate the state-action pairs not in the dataset. Please read the BCQ [3] paper (pioneering work in offline RL) carefully again to understand the main problem in Offline RL.
   1. This paper claims that BCQ [3] paper proposes the major challenge of "distribution shift between learned policy and the behavior policy", which necessitates behavior regularization in line 53 page 1 and line 88 page 2.
   2. However, the central claims of BCQ is to address the OOD-action problem since exploration is not allowed in offline setting.
   3. The concept of "distributional shift" is the main concern in Offline to Online (O2O) RL rather than Offline RL [4].
      1. The prior trained policy with offline data should be continually trained online.
      2. Offline data and online data may have "distributional shift".

2. The writing lacks logic. For example,
   1. the last sentence in Offline RL has little relationship with the form writing. Why do you jump to it directly?
   2. why do you mention model-based RL in Safe RL? Is it an important part in this paper or related to QPT? Model-based RL can be applied in any RL fields.
   3. what is the meaning of "zero constraint violation"? It can be explained as "Constrained problem" ($C(\tau)\leq k$) or "Persistantly Safe problem" ($C(\tau)=0$). This confusion is related to your experiment in the following.

[1] Lin, Q., Tang, B., Wu, Z., Yu, C., Mao, S., Xie, Q., ... & Wang, D. (2023, July). Safe offline reinforcement learning with real-time budget constraints. In *International Conference on Machine Learning* (pp. 21127-21152). PMLR.

[2] Yao, Y., Cen, Z., Ding, W., Lin, H., Liu, S., Zhang, T., ... & Zhao, D. (2024). Oasis: Conditional distribution shaping for offline safe reinforcement learning. *Advances in Neural Information Processing Systems*, *37*, 78451-78478.

[3] Scott Fujimoto, David Meger, and Doina Precup. Off-policy deep reinforcement learning without exploration. In *Proc. of the International Conference on Machine Learning*, Long Beach, CA, June 2019.

[4] Uchendu, I., Xiao, T., Lu, Y., Zhu, B., Yan, M., Simon, J., ... & Hausman, K. (2023, July). Jump-start reinforcement learning. In *International Conference on Machine Learning* (pp. 34556-34583). PMLR.

## Weakness on Experiment

### 1. The baselines are not updated and suitable.

1. It is not to ask QPT to compare all other baselines. However, the baselines are mainly from 2019-2023.

2. FISOR is published in 2024 but it focuses on a different problem.

Newest and suitable baselines include:

1. Methods adapted to various constraints without retraining
   1. CAPS [5], 2025 AAAI
   2. CCAC [6], 2025 ICLR
2. Methods with other data augmentation methods:
   1. OASIS [2], 2024 NIPS
   2. TREBI [1], 2023 ICML

### 2. The cost threshold is 1 in Table 1 and Figure 1 while Table 4 describes cost threshold as 10. Is there any problem?

### 3. The constraint selection are not aligned with the problem.

1. We hope the author to clarify the differences between "Constrained Problem" and "Persistantly Safe (or Zero Cost) Problem".
2. This paper focuses on Constrained problem (in problem setup) while the main results focus on "zero cost", which set cost threshold to 1.
3. This setting may follow FISOR's setting. However, FISOR focuses on "zero cost problem".
4. To be focused on the constrained problem, most existing paper selects three thresholds (better in different ranges, such as 10, 20, 40) and test average performance.

### 4. Hyper parameter anaylsis

The hyper-parameters ($\eta_1$ and $\eta_2$) lacks general applicability in Figure 1. For example, $\eta_1\leq0.01$ is suitable for SwimmerVelocity but is not for HopperVelocity when $\eta_2=0.01$.

1. Does the results on Table 1 follow the same $\eta_1$ and $\eta_2$?
2. Or is it specially finetuned for each task?
3. Please list the value in Table 4 no matter what case.

### 5. Ablation of different component

1. The data augmentation is not the contribution of this paper. Why do you introduce it in Table 2?

### 6. Ablation of zero-shot adaptation

For ablation of zero-shot adaptation, the cost range a little bit narrow (from 1 to 6). In the offline dataset, The cost ranges of Velocity task are from 0 to 200, 250, or even 300. The cost range utilized in the ablation is to narrow to reflect the adaptive ability to different costs without retraining.

[5] Chemingui, Y., Deshwal, A., Wei, H., Fern, A., & Doppa, J. (2025, April). Constraint-adaptive policy switching for offline safe reinforcement learning. In *Proceedings of the AAAI Conference on Artificial Intelligence* (Vol. 39, No. 15, pp. 15722-15730).

[6] Guo, Z., Zhou, W., Wang, S., & Li, W. (2025, March). Constraint-conditioned actor-critic for offline safe reinforcement learning. In *The Thirteenth International Conference on Learning Representations*.



## Confusion on Theoretical analysis

### 1. Notations are not explained clearly

1. In different RL fields, similar notations may represent a wide range of meanings. It is better to explain it before used in the paper.
2. If some notations are from another paper, please also introduce the notations.

### 2. Binary assumption

1. Most rewards do not satisfy the binary rewards assumption.
2. While the costs are binary in tested benchmarks, costs can also be continuous value.

### 3. Analysis result
1. This paper focus on Constrained MDP, where the costs only need to satisfy the constraint rather than minimization.
   1. However, this paper only provides the probabilities of selecting actions with maximum reward or minimum cost.
   2. The analysis on the probabilities of constraint satisfaction is not provided.


2. In Eq (8), the loss function or the objective does not include any constraint satisfaction loss, but only focus on reward maximization and cost minimization.
   1. If reward maximization and cost minimization are conflicting with each other, how to guarantee that QPT will first guarantee constraint satisfaction?
3. The weight of reward maximization and cost minimization loss is controlled by $\eta_1$ and $\eta_2$, which are predefined.
   1. Given the theoretical analysis, how to determine the suitable value before testing online since offline RL does not allowing any online finetuning.
   2. Why are the hyperparameters still need to be manually set?

**Questions:**

Please see the weakness section.
Besides, it is suggested that the author can rewrite this paper following a good article structure and clearly explain the central claim, content from previous work, and the key idea of QPT.
1. Please write the related work in logic rather than list all papers, no matter it is related to this paper or not.
2. Please clearly classify the content of the previous work and the improvement in your paper.
    1. It is very important! In severe cases, **it may lead to academic plagiarism**.

---

### Official Review · Reviewer_tdM2 · 2025-10-30

**Soundness:** 2
**Presentation:** 2
**Contribution:** 2
**Rating:** 4
**Confidence:** 3

**Summary:**

This paper proposes the Q-learning Penalized Transformer (QPT), an algorithm for offline safe reinforcement learning. QPT integrates Transformer-based sequence modeling with Q-learning penalization to balance safety, reward maximization, and behavior regularization. It learns action distributions conditioned on historical trajectories and target returns while optimizing both reward and cost Q-networks. The method enables zero-shot adaptation to varying safety thresholds without retraining. Theoretical analysis proves convergence toward near-optimal safe policies, and experiments on benchmark tasks demonstrate that QPT outperforms several baseline methods.

**Strengths:**

+ The paper is generally well-written and easy to follow. In addition to empirical studies, the authors also provide a theoretical analysis of the proposed algorithm.

**Weaknesses:**

- The paper mainly combines existing techniques without introducing substantial algorithmic novelty. The core components, Decision Transformer, Q-learning penalization, and data augmentation, are all established methods; this work primarily reassembles them within a single framework.

-  The discussion of related work is insufficient. In the domain of offline safe reinforcement learning, many recent studies [1, 2, 3, 4] have been proposed, yet they are neither discussed nor compared in this paper.

- Although theoretical analysis is provided, the assumptions are overly strong. The return coverage assumption in Theorem 3.1 requires the offline dataset to have a great amount of coverage, even including all unsafe state-action pairs, this is almost impossible to satisfy in reality.

- The set of baselines in the main experiments is limited; additional competitive algorithms should be included for a more comprehensive evaluation.

[1] Yihang Yao, Zhepeng Cen, Wenhao Ding, Haohong Lin, Shiqi Liu, Tingnan Zhang, Wenhao Yu, and Ding Zhao. Oasis: Conditional distribution shaping for offline safe reinforcement learning. Advances in Neural Information Processing Systems, 37:78451–78478, 2024.

[2] Yassine Chemingui, Aryan Deshwal, Honghao Wei, Alan Fern, and Jana Doppa. Constraint-adaptive policy switching for offline safe reinforcement learning. In Proceedings of the AAAI Conference on Artificial Intelligence, volume 39, pages 15722–15730, 2025.

[3] Ze Gong, Akshat Kumar, and Pradeep Varakantham. Offline safe reinforcement learning using trajectory classification. In Proceedings of the AAAI Conference on Artificial Intelligence, volume 39, pages 16880–16887, 2025.

[4] Woosung Kim, JunHo Seo, Jongmin Lee, and Byung-Jun Lee. Semi-gradient dice for offline constrained reinforcement learning. arXiv preprint arXiv:2506.08644, 2025.

**Questions:**

1) In Theorem 3.1, the authors consider an MDP with binary rewards and costs. Does this refer to {0, 1}-valued reward and cost signals? If so, is this simplification too restrictive?

2) In Equation (8), the coefficients $\eta_1$ and $\eta_2$ are fixed. Could they be adaptively updated based on constraint violations, as in Lagrangian methods, using multiplier updates?

---

### Official Review · Reviewer_nRie · 2025-11-01

**Soundness:** 3
**Presentation:** 3
**Contribution:** 3
**Rating:** 4
**Confidence:** 3

**Summary:**

The paper presents Q-learning Penalized Transformer (QPT), a new framework for safe offline reinforcement learning that seeks to maximize reward while meeting safety constraints and staying consistent with the offline behavior policy. QPT integrates a conditional Transformer policy—which models trajectories conditioned on target return and cost—with Q-learning-based penalization, using separate reward and cost Q-networks to guide learning. It further enhances safety and generalization through data augmentation for infeasible targets and an ensemble inference scheme that selects safe, high-reward actions during testing. Under mild theoretical assumptions, QPT is shown to approximate near-optimal policies, and experiments on 38 benchmark tasks demonstrate clear performance gains over prior safe offline RL methods, especially in zero-shot adaptation to unseen safety constraints.

**Strengths:**

- This paper tackles the Safe RL problem, addressing the central challenge of simultaneously achieving safety constraint satisfaction, reward maximization, and behavior regularization. The explicit emphasis on behavior regularization is particularly noteworthy, as it strengthens policy robustness in offline settings—an aspect that is often underexplored in prior safe RL research.
- From a theoretical perspective, the paper establishes formal guarantees demonstrating that the proposed QPT framework can recover near-optimal safe policies under mild conditions, including return coverage and near-deterministic dynamics. It further derives theoretical bounds that characterize the performance gap between the learned and optimal policies in both reward and cost objectives, providing solid justification for the framework’s design and underlying principles.
- The paper provides comprehensive experimental evaluations across multiple benchmark suites and a variety of strong baselines. The results consistently show that QPT surpasses existing methods across diverse environments, underscoring its effectiveness, robustness, and strong generalization capability in safe offline reinforcement learning.
- The paper is clearly written and well organized, moving smoothly from the motivation to the method and then to the experiments, making it easy for readers to follow the main ideas and technical details.

**Weaknesses:**

- Although the paper introduces the Q-Learning Penalized Transformer (QPT), the approach mainly extends existing ideas by combining the Constrained Decision Transformer (CDT) with Q-learning penalization. Since prior studies [1–3] have already explored similar ways of integrating Q-learning regularization into Transformer-based policies, QPT comes across as a modest refinement rather than a major conceptual breakthrough.
- The experimental section does not include an analysis of runtime efficiency or computational complexity, which is particularly relevant since QPT combines a Transformer-based policy with multiple Q-networks. This architecture likely introduces significant training and inference overhead, yet the paper offers no empirical results or discussion on its scalability or computational cost.
- Table 1 presents the performance of various algorithms across different environment settings; however, it lacks the reporting of standard deviations or confidence intervals, which are essential for assessing the statistical reliability and robustness of the results.
- This paper lacks comparisons with several recent and strong baselines (e.g., [4–5]), which makes it difficult to fully assess the claimed performance improvements and the novelty of the proposed method relative to the current state of the art.
- The ablation study varies only η₁ and η₂, without exploring other critical factors such as sequence length, n-step horizon, ensemble size, or normalization constants. Consequently, the stability of the model under varying hyperparameter settings is not well understood.
- In Figure 2, CDT frequently violates the cost constraints, especially in Walker2dVelocity, CarButton1, and PointButton1. This outcome contradicts the results reported in [6], where CDT demonstrated better safety performance. The inconsistency may arise from differences in experimental configurations or hyperparameter choices. It would be helpful for the paper to include results obtained under the same settings as [6] and briefly explain the source of this discrepancy.

**Questions:**

- Could you provide the standard error of the results in Table 1 for clarity?
- Could you please add the performance comparison with recent baselines such as [2], [3], [4], and [5]?
- Could you please use the same environment settings as in [6] to reproduce the results shown in Figure 2?
- Could you please show whether QPT is robust to conflicting reward returns? Like the Figure 5 shown in the [6].
- Could you please use different environment settings to conduct the ablation studies and include more ablation results?
- What is the computational overhead (e.g., training time and GPU memory usage) of QPT compared to the baseline methods?

## References
- **[1]** Yamagata, Taku, Ahmed Khalil, and Raul Santos-Rodriguez. "Q-learning decision transformer: Leveraging dynamic programming for conditional sequence modelling in offline rl." International Conference on Machine Learning. PMLR, 2023.

- **[2]** Wang, Ruhan, and Dongruo Zhou. "Safe Decision Transformer with Learning-based Constraints." 7th Annual Learning for Dynamics\& Control Conference. PMLR, 2025.

- **[3]** Hu, Shengchao, et al. "Q-value regularized transformer for offline reinforcement learning." arXiv preprint arXiv:2405.17098 (2024).

- **[4]** Guan, Jiayi, et al. "Voce: Variational optimization with conservative estimation for offline safe reinforcement learning." Advances in Neural Information Processing Systems 36 (2023): 33758-33780.

- **[5]** Wei, Honghao, et al. "Adversarially trained weighted actor-critic for safe offline reinforcement learning." Advances in Neural Information Processing Systems 37 (2024): 52806-52835.

- **[6]** Liu, Zuxin, et al. "Constrained decision transformer for offline safe reinforcement learning." International conference on machine learning. PMLR, 2023.

---

### Meta-Review · Area_Chair_JDUF · 2025-12-08

**Summary:**

**Summary of the Paper**:
This paper introduces a Q-learning penalized transformer (QPT) for developing a safe offline RL policy. In particular, this paper considers reward maximization, constraint satisfaction, and behavioral policy regularization in their objective. The paper introduces a conditional transformer structure to train the policy. The empirical results show their strength compared to the baselines.

**AC's overview**: While the paper's developed methodology is based on sound theory, many reviewers raised concerns about the novelty of the proposed contributions.

(1) The reviewers' main concerns are that there are existing approaches to solve offline CMDP; however, the paper did not compare with the recent ones. For example, the reviewer nRie provided a bunch of references. I agree on that assessment. Also, it is not clear whether the technical novelties of the work are compared to the conditional decision-transformer, as this paper also relies on data augmentation.

(2) As the reviewer tdM2 pointed out this paper requires a strong coverage assumption. In particular, it seems that (9) , and (10) requires that the data have coverage even for unsafe state-action pairs, which is not practical.

(3) Several reviewers also pointed out the assumptions used for showing the theoretical results. In particular, they used a binary cost assumption. Also, it is not clear how to tune the values $\eta_1$, and $\eta_2$.

**Reviewer Concerns:**

Reviewer concerns have not been addressed as the authors did not provide any rebuttal.

**Reviewer Scores:**

N/A

The authors did not provide any rebuttal.

---

### Decision · Program_Chairs · 2026-01-26

Reject